# Transcriptome and Metabolome Analyses Reveal Perfluorooctanoic Acid-Induced Kidney Injury by Interfering with PPAR Signaling Pathway

**DOI:** 10.3390/ijms241411503

**Published:** 2023-07-15

**Authors:** Yan Zhang, Yang Li, Nana Gao, Yinglan Gong, Wanyu Shi, Xiaodan Wang

**Affiliations:** 1College of Traditional Chinese Veterinary Medicine, Hebei Agricultural University, Baoding 071001, China; zhangyan2333333@163.com (Y.Z.); gaonanahbnd2023@163.com (N.G.); 13803269157@126.com (W.S.); 2Veterinary Biological Technology Innovation Center of Hebei Province, College of Traditional Chinese Veterinary Medicine, Hebei Agricultural University, Baoding 071001, China

**Keywords:** perfluorooctanoic acid, offspring mice, kidney injury, transcriptomics, metabolomics

## Abstract

Perfluorooctanoic acid (PFOA) is widely used in aviation science and technology, transportation, electronics, kitchenware, and other household products. It is stable in the environment and has potential nephrotoxicity. To investigate the effect of PFOA exposure during pregnancy on the kidneys of offspring mice, a total of 20 mice at day 0 of gestation were randomly divided into two groups (10 mice in each group), and each group was administered 0.2 mL of PFOA at a dose of 3.5 mg/kg or deionized water by gavage during gestation. The kidney weight, kidney index, histopathological observation, serum biochemistry, transcriptomics, and metabolomics of the kidneys of the 35-day offspring mice were analyzed. In addition, malondialdehyde (MDA), superoxide dismutase (SOD), and catalase (CAT) levels in the kidney were measured. Transcriptome analysis results showed that 387 genes were up-regulated and 283 genes were down-regulated compared with the control group. These differentially expressed genes (DEGs) were mainly concentrated in the peroxisome-proliferator-activated receptor (PPAR) signaling pathway and circadian rhythm. Compared with the control group, 64 and 73 metabolites were up- and down-regulated, respectively, in the PFOA group. The altered metabolites were mainly enriched in the biosynthesis of unsaturated fatty acids. PFOA can affect the expression levels of circadian rhythm-related genes in the kidneys of offspring mice, and this change is influenced by the PPAR signaling pathway. PFOA causes oxidative stress in the kidneys, which is responsible for significant changes in metabolites associated with the biosynthesis of unsaturated fatty acids.

## 1. Introduction

Per- and polyfluoroalkyl substances (PFAS) are a class of artificially synthesized organic fluorine compounds, which are widely used in civil and industrial products due to their excellent stability and hydrophobic and oleophobic properties [1,2]. PFAS was first detected in the blood of occupational exposed workers in the 1970s and the general population in the 1990s, making people aware of its potential health risks [3]. Although the production of PFAS began to be phased out in the early 2000s, the United States Centers for Disease Control and Prevention still reported that PFAS was detected in the blood of almost all American adults and accumulated in body tissues, indicating their persistence in the biological system. Because of their widespread use and a half-life of 4 to 10 years of PFAS, they were detected in various natural environments, such as water, air, soil, wildlife, and even in remote polar regions [4,5,6]. Perfluorooctanoic acid (PFOA), a kind of PFAS, is the most stable transformation product of perfluorinated precursors and derivatives in the environment. In 2014, the International Agency for Research on Cancer (IARC) classified PFOA as a possible human carcinogen (group 2B) based on some epidemiological evidence related to kidney cancer (IARC, 2007). In people with high exposure to PFOA, the serum concentration of PFOA is higher, which is correlated with kidney cancer, and the cancer mortality rate is increased [7,8]. Because of its hydro- and oleophobic properties, PFOA does not accumulate in adipose tissue. It is primarily stored in the blood, muscle, testis, liver, and kidney [9]. PFOA can cause liver damage [10] and reproductive damage [11]. Kidney disease is associated with exposure to environmental pollutants, such as the common surfactant PFOA in various consumer products [12]. The kidney is the main metabolic organ of both human and animal bodies and one of the main target organs of PFOA toxicity [13]. However, only a few studies have examined the toxic effects of PFOA on kidney function.

Transcriptomics studies can help understand the process of gene expression under defined conditions to examine the function of genes exhibiting altered expression. Transcriptome information has been used increasingly to understand the toxic mechanisms of environmental pollutants in organisms. Exposure to PFOA can enhance intracellular oxidative stress in rat primary liver cells. Full-length transcriptome analysis shows that 100 μM PFOA exposure for 24 h can significantly change the transcription of 177 genes [14]. Hexafluoropropylene oxide trimer acid (HFPO-TA), a substitute for PFOA, causes liver damage in mice, including hepatomegaly, necrosis, and increased alanine aminotransferase activity. In addition, the total cholesterol and triglycerides decreased in a dose-dependent manner in the liver. Liver transcriptome analysis has shown that the expression levels of 2491 genes in the mouse liver differ from those in the sham-exposed controls. The expression levels of proteins associated with carcinogenesis, such as alpha-1-antiproteinase F (AFP), P21, sirtuin 1 (Sirtl), C-MYC, and proliferating cell nuclear antigen (PCNA), were significantly elevated [15].

Metabolomics enables dynamic monitoring of compounds in the body caused by internal and external factors, which can be used to evaluate the effects of environmental pollutants [16,17] and drugs [18] and infer their potential mechanisms of action. Levels of phenylalanine, tyrosine, and tryptophan are significantly affected when the nematodes are exposed to PFOA and perfluorooctane sulfonate (PFOS). At the same time, phospholipid and triacylglycerol are disturbed in the exposed group [19]. Kyoto Encyclopedia of Genes and Genomes (KEGG) annotation has emphasized that PFOA treatment can interfere with amino acid carbohydrate and lipid metabolism, resulting in immune system disorders [20]. In diabetic patients, PFOA and PFOS exposure aggravated diabetic kidney injury by disrupting the metabolism of amino acids and purines [21].

Therefore, this study aimed to explore the kidney damage caused by pregnancy exposure to PFOA in mice offspring and gain insights into the possible mechanism of toxicity at the molecular level via transcriptomics and metabolomics analyses.

## 2. Results

### 2.1. Effects of PFOA Exposure on Kidney Weight, Kidney Index and Organ Structure

After exposure to PFOA during pregnancy, the kidney weight of offspring mice in the PFOA group decreased (*p* < 0.05), and the kidney index decreased significantly (*p* < 0.01) relative to the control group (Figure 1A,B). Histopathological observation showed that the kidney structure was clear, and the kidney tubules were arranged in an ordered manner in the control group. However, the kidney tubules of the PFOA group were disorganized, and the kidney tubular structure was diminished in size (Figure 1C).

### 2.2. Effects of on Serum Creatinine (CRE) and Blood Urea Nitrogen (BUN) and Markers of Serum Kidney Injury

As shown in Figure 2A,B, compared with the control group, 3.5 mg/kg body weight (BW) PFOA significantly increased CRE and BUN levels of mice. Furthermore, PFOA exposure significantly increased (*p* < 0.01) the levels of Cystatin C (Cys-C), kidney injury molecule 1 (KIM-1), and neutrophil gelati-nase-associated lipocalin (NGAL) in offspring mice (Figure 2C–E) compared with the control group.

### 2.3. Transcriptome Analysis of Kidney

Transcriptome results showed that a total of 670 DEGs were identified by comparing the PFOA treatment group with the control group. Of these, 387 and 283 genes were up-and down-regulated in the PFOA group (Figure 3A). Figure 3B shows the heat map of differential gene expression between the control and PFOA groups. The expression levels of different genes between the two groups could be observed.

As is shown in Figure 3C, DEGs were mainly involved in circadian rhythm, the rhythmic process, the steroid metabolic process, the alcohol metabolic process, organic acid binding, and other biological processes. Then, the KEGG database was used for pathway enrichment analysis (Figure 3D).

### 2.4. Quantitative Real-Time Polymerase Chain Reaction (qRT-PCR) of DEGs

The circadian-rhythm-related genes from the results of transcriptome sequencing were selected for analysis by qRT-PCR. As shown in Figure 4A–E, expression levels of per2, rev-erbα, dec, bmal1, and clock mRNA showed a strong correlation with the data obtained through RNA-seq, validating the reliability of the transcriptomic results.

The mRNA levels of *per2*, *rev-erbα*, and *dec* were decreased in the PFOA group, while the expression of *bmal1* was significantly increased. Interestingly, *clock* expression was unchanged in the PFOA group compared to the control group.

### 2.5. Metabolomics Analysis of Kidney

To verify the effect of PFOA exposure during pregnancy on renal metabolites of offspring mice, liquid chromatograph mass spectrometry (LC-MS) was used to conduct untargeted metabolomics analysis on kidney tissues. To avoid interference from external factors, it was necessary to control the data quality. Therefore, the correlation coefficient between QC samples was calculated. The higher the correlation of QC samples, the better the stability of the whole test process. The correlation analysis of QC samples in positive and negative modes is shown in Figure 5A,B. R2 was close to 1, which indicates that the test data are reliable.

Partial least squares discrimination analysis (PLS-DA) results showed significant separation between the control and PFOA groups (Figure 5C). To test whether the PLS-DA model was “over-fitting”, the model was sorted for validation. The sorting verification results (Figure 5D) showed that R2 > Q2, the intercept between the Q2 regression line and Y-axis, was less than 0, which indicates that the model is not “overfitted”. The lack of “over-fitting” indicates that the model can describe the samples well and can be used to search for model biomarker groups. The volcano map presented in Figure 5E is based on the metabolites detected in the analysis. A total of 147 metabolites were detected, of which 72 were up-regulated, and 75 were down-regulated. Figure 5F is a cluster heat map based on all the altered metabolites.

Analysis of all annotated altered metabolites by KEGG enrichment showed that the main pathways involved in these altered metabolites included the biosynthesis of unsaturated fatty acids, vitamin digestion and absorption, pyrimidine metabolism, fatty acid biosynthesis, and ABC transporters (Figure 6A). Among these, the biosynthesis of unsaturated fatty acids had the strongest correlation, which enriched altered metabolites, including palmitic acid, docosanoic acid, stearic acid, and docosapentaenoic acid. Compared with the control group, the relative concentrations of palmitic and docosapentaenoic acids significantly increased and decreased, respectively. PFOA was the most significantly altered metabolite, with significantly higher PFOA levels observed in the PFOA-exposed group (Figure 6B–F). As shown in Figure 6G–I, PFOA exposure significantly decreased (*p* < 0.05) the activities of catalase (CAT) and superoxide dismutase (SOD) in kidney tissue. The malondialdehyde (MDA) levels increased (*p* < 0.01) in the PFOA-exposed group.

## 3. Discussion

Gestation is the period during which embryos and later fetuses mature in the matrix. The kidney is the main metabolic organ of both human and animal bodies and one of the main target organs of PFOA [13]. A previous study determined toxic doses of PFOA and found that exposure to 3.5 mg/kg during pregnancy had a significant effect on offspring [22].

Creatinine is formed in muscle from creatine phosphate by spontaneous and irreversible transformation, which is released into the blood. Urea nitrogen is the product of protein catabolism. The majority of these substances are excreted by the kidney. Following damage to the kidneys, the levels of creatinine and urea nitrogen in the blood increase [23,24]. To determine the harmful effect of 3.5 mg/kg PFOA on the kidney, concentrations of Cys-C, KIM-1, and NGAL and markers of kidney injury were measured. Cys-c is a non-glycosylated protein that can be expressed in all nucleated cells and is an endogenous marker of mammals. The kidney is the only scavenging organ for Cys-c and accurately reflects the glomerular filtration rate. Elevation of Cys-c levels suggests impaired kidney function [25]. KIM-1 is highly expressed in the proximal kidney tubular epithelial tissue and can be detected in the blood and urine of patients experiencing acute kidney tubular injury [26,27].

NGAL is a low-molecular-weight protein secreted by human neutrophils. The protein is under-expressed in most tissues, but protein levels increase dramatically when the kidney is injured. NGAL is the most valuable marker for the early diagnosis of kidney injury [28]. The data presented in this study showed that compared to the control group, 3.5 mg/kg PFOA significantly increased the serum concentrations of Cys-C, KIM-1, and NGAL in offspring mice. Therefore, exposure to 3.5 mg/kg PFOA during pregnancy can damage the kidneys of offspring mice.

To explore the mechanism of kidney damage in offspring mice exposed to PFOA during pregnancy, renal transcriptome analysis was performed. Transcriptome results showed that, compared with the control group, 387 genes were up-regulated, and 283 genes were down-regulated. Therefore, the genes are considered potential biomarkers for PFOA-induced kidney injury in offspring mice. However, it is not yet clear whether these genes are responsible for the development of toxicity. The results obtained from the GO enrichment analysis showed that the DEGs were mainly involved in circadian rhythm, the rhythmic process, the functional metabolic process, the alcohol metabolic process, organic acid binding, and other biological processes. The results of the KEGG analysis showed that DEGs were mainly enriched in the peroxisome-proliferator-activated receptor (PPAR) signaling pathway and circadian rhythm. PFOA can act as a ligand of PPARα to activate the PPARα pathway and induce injury in mice [29]. The administration of 3.0 mg/kg PFOA to wild-type mice and PPARα knockout mice for 7 consecutive days was recently reported. Peroxisome and mitochondrial β-oxidation associated genes in PPARα knockout mice were more significant. Therefore, PFOA-induced fatty acid metabolism abnormalities in mice are regulated by PPARα [30]. PFOA exposure can result in liver injury in offspring mice through altered expression of the PPARα signaling pathway [10], which is consistent with our transcriptome data.

In addition, another pathway was significantly enriched in the KEGG data: circadian rhythm. Circadian rhythm refers to physiological processes that occur in cycles of approximately 24 h. This phenomenon is mainly regulated by circadian genes [31]. The *clock* and *bmal1* genes are the core genes of the circadian rhythm. These genes can form heterodimer complexes in the cytoplasm, which are translocated to the nucleus and bind to e-box enhancer sequences to activate or repress other core clock genes, including per, dec, rev-erbα. However, the protein products of *per*, *dec*, and *rev-erbα* can inhibit the transcription of *clock* and *bmal*1, forming a feedback loop [32,33]. The expression levels of three subtypes of PPAR are involved in the regulation of the circadian rhythm through direct regulation of the expression of core genes of the biological clock [34]. Furthermore, e-box enhancer binding of bmal1 and clock heterodimers is regulated by PPARα [35] (Figure 7). Grimaldi reported that *per2* inhibited PPARγ, and the per2-deficient mice exhibited altered lipid metabolism [36]. Some other environmental pollutants, such as dibutyl phthalate (DBP), can affect the expression of the circadian-rhythm-related gene (*Bmal1*) in *Drosophila melanogaster*, resulting in circadian disruption [37]. In addition, marine medaka *(Oryzias melastigma)* exposed to benzo[a] pyrene (BaP) exhibited significantly altered expression levels of circadian-rhythm-related genes such as *per1* and *cry1*, and even the toxic effects can be transferred to the next generation [38]. The data presented showed that, compared with the control group, 3.5 mg/kg PFOA significantly up-regulated the expression levels of *per*, *dec*, and *rev-erbα* and significantly down-regulated the expression of *bmal1*. The expression of the *clock* was unchanged, which is consistent with the transcriptome results.

Metabolites are the final products of life activities, and their changes can better reveal whether the organism is affected by PFOA. Our metabolome results showed that 147 metabolites were altered, with 72 up-regulated and 75 down-regulated metabolites, respectively. It is noteworthy that PFOA increased in the kidneys of offspring mice exposed to 3.5 mg/kg PFOA during pregnancy, which suggests that the kidney is a target organ of PFOA, and PFOA exposure can occur by crossing the placental or blood–milk barriers. In addition, the relative concentration of palmitic acid was significantly increased, and the relative content of docosapentaenoic acid was significantly decreased. Palmitic acid is the main constituent of fatty acids in the body. Palmitic acid leads to oxidative stress in the kidney tubular epithelial cells in humans, which results in renal insufficiency [39]. Our data demonstrated that exposure to 3.5 mg/kg PFOA significantly increased the renal levels of palmitic acid in offspring mice. The pathway enrichment analysis of differentially expressed metabolites showed that exposure to 3.5 mg/kg PFOA during pregnancy affected the biosynthesis of unsaturated fatty acids in the kidneys of offspring mice. Fatty acids are an intermediate product of lipid metabolism, which can be sensitive to lipid metabolism. According to the degree of saturation of carbon-hydrogen (C-H) bonds, fatty acids can be divided into saturated and unsaturated fatty acids. Yang et al. reported that PFOA exposure significantly reduced the ratio of unsaturated fatty acids to saturated fatty acids in *Escherichia coli* cells [40], which is due to the mass production of reactive oxygen species (ROS), which reacts with the unsaturated fatty acids on the membrane to produce lipid peroxidation, thus reducing the content of unsaturated fatty acids.

To demonstrate that exposure to PFOA during pregnancy can lead to kidney oxidative stress in offspring mice, levels of MDA, a lipid peroxidation product, were detected, which can reflect the degree of lipid peroxidation in the body. Furthermore, SOD and CAT, which are necessary antioxidant enzymes in the body, were also analyzed [41,42]. The data suggest that 3.5 mg/kg PFOA can significantly increase the level of MDA and significantly decrease the levels of SOD and CAT in the kidney of offspring mice. Therefore, we suggest that PFOA induces the production of a large number of ROS in the kidney, which attacks the unsaturated fatty acids on the biofilm, leading to lipid peroxidation and the production of a large amount of MDA.

## 4. Materials and Methods

### 4.1. Animals and Experimental Design

Kunming mice (8 weeks of age), including females and males, were provided by SPF Biotechnology Co., Ltd. (Beijing, China). The license number was SCXK (Beijing, China) 2019-0010. The animals were provided food and water ad libitum and housed in the vivarium at the College of Veterinary Medicine, Hebei Agricultural University, China. Ambient temperature was maintained at 21 ± 1 °C, with a light/dark cycle of 12/12 h. After one week of acclimation, male and female mice were caged at a ratio of 2:1 in the evening. The next day, female mice were examined for vaginal plugs and were considered day 0 of pregnancy. Twenty pregnant mice were randomized into two groups, including control (*n* = 10 mice) and PFOA (*n* = 10 mice) groups. Then, 0.2 mL of ultrapure water and 3.5 mg/kg of PFOA (98% purity, Sigma, St. Louis, MO, USA) were administered orally every morning from day 1 to day 17 of gestation. After birth, the offspring were conventionally housed with their mother until weaning on day 21. On day 35 after its birth, the offspring mice (*n* = 10) were weighed and given euthanasia through cervical dislocation to reduce the pain of animals. Kidney samples were aseptically collected and weighed. The kidney index was calculated as the ratio of kidney weight (g) to body weight (g). The left kidneys were fixed in 4% paraformaldehyde, and all right kidneys were flash-frozen in liquid nitrogen and stored at −80 °C for subsequent use. At the same time, fresh blood was placed at room temperature for 30 min, centrifuged at 3500× *g* for 15 min to separate the serum, and stored at −20 °C for further use.

### 4.2. Kidney Histopathological Observation

Kidney tissues fixed in 4% paraformaldehyde were dehydrated, clarified, infiltrated, and embedded in wax blocks using standard histological methods. The blocks were then sectioned, mounted on slides, and stained with hematoxylin and eosin (H&E) for observation on a light microscope (Olympus Corporation, Tokyo, Japan).

### 4.3. Determination of CRE and BUN in Serum

CRE and BUN levels were measured in the offspring of mice using the Creatinine Assay kit (sarcosine oxidase) and Urea Assay kit (Nanjing Jiancheng Technology Co., Ltd., Nanjing, China), according to the manufacturer’s instructions.

### 4.4. Determination of Kidney Damage Markers in Serum

Cys-C, KIM-1, and NGAL levels in serum were measured using commercially available enzyme-linked immunosorbent assay (ELISA) kits (Shanghai Enzyme-linked Biotechnology Co., Ltd., Shanghai, China), according to the manufacturer’s instructions.

### 4.5. Transcriptomic Analysis

Total RNA was extracted from the kidneys of 3 randomly selected from each group. RNA integrity was assessed on Agilent 2100 BioAnalyzer (Agilent Technologies, Santa Clara, CA, USA). The NEBNext^®^ Ultra™ RNA Library Prep Kit for Illumina^®^ (NEB, Ipswich, MA, USA) was used to construct the RNA library. Illumina sequencing was performed after a qualified library inspection on the Agilent Bioanalyzer 2100 system (Agilent Technologies, Santa Clara, CA, USA). Sequencing reactions were conducted by Beijing Novogene Biological Information Technology Co., Ltd. (Beijing, China). After completing gene expression quantification, DEGs were analyzed in three main steps: normalizing the original read count, calculating the hypothesis testing probability for the statistical model, and performing multiple hypothesis testing and correction to obtain the false discovery rate (FDR). The screening criteria for DEGs are|log2 (fold change)| > 1 & padj <= 0.05.

### 4.6. qRT-PCR Results

Total RNA was extracted from kidney tissue according to the instructions of the total RNA extraction kit (Beijing Promega Biotech Co., Ltd., Beijing, China). Total RNA was transcribed into complementary DNA (cDNA) using the HiScript^®^ Ⅲ RT SuperMix for qPCR kit (Nanjing Vazyme Hiotech Co., Ltd., Nanjing, China). Then, the cDNA was amplified by qPCR using the TB Green^®^ Premix DimerEraserTM (Takara, Dalian, China) on the Light Cycler 480 qRT-PCR system (Light Cycler 480, Roche, IN, USA). Primer design and synthesis were performed by Takara (Dalian, China). Primer sequences are presented in Table 1. Using β-actin as an internal reference gene, the relative expression level of each gene was calculated according to the 2^−∆∆Ct^ method.

### 4.7. Metabonomics Analysis

Liquid chromatography–tandem mass spectrometry (LC-MS) was used to study untargeted metabolomics. For metabolic analysis of kidney tissue, metabolites were extracted. The kidney sample was milled with 100 mg liquid nitrogen and placed in a PE tube, and 500 μL 80% methanol was added to each sample. After vortex oscillation, the samples were incubated on ice for 5 min and centrifuged at 15,000× *g* at 4 °C for 20 min. The supernatant was diluted with ultrapure water until the methanol content was 53% and centrifuged at 15,000× *g* at 4 °C for 20 min. The supernatant was collected and set aside. LC-MS was used for both qualitative and quantitative analyses, and the functions and classification of identified metabolites were annotated by Beijing Novogene Biological Information Technology Co., Ltd. (Beijing, China). Differential metabolites were screened, and a threshold of fold change (FC) > 1.2 or FC < 0.833 and *p* value < 0.05 was set.

### 4.8. Determination of MDA, CAT, SOD

Kidney samples were diluted with normal saline at a ratio of 1:9 for a 10% tissue homogenate. The homogenate protein concentration was measured and calculated using a protein quantification kit. The levels of SOD, MDA, and CAT in the kidney tissues of each group were measured according to the instructions of the kit. All kits were purchased from Nanjing Jiancheng Technology Co., Ltd., Nanjing, China.

### 4.9. Statistical Analyses

The experimental results were conducted using Excel, and IBM SPSS v.19.0 was used to conduct an independent sample *t*-test on the data. Differences were considered statistically significant at *p* < 0.05 and extremely significant at *p* < 0.01. All results were expressed as mean ± standard deviation (x ± SD). The histogram was drawn using GraphPad Prism 9.0 (GraphPad Software, San Diego, CA, USA). The transcriptomics and metabolomic data analyses and graphics were performed using novomagic 1.0.7 (Novogene, Beijing, China).

## 5. Conclusions

In this study, exposure to PFOA during pregnancy resulted in kidney damage in offspring mice. Our research accounts for the significant changes in metabolites associated with the biosynthesis of unsaturated fatty acids. Additionally, PFOA affects the PPAR signaling pathway and the circadian rhythm. These two pathways interact to influence kidney function in offspring mice. The research will enrich the study on mechanism of PFOA damage repair.

## Figures and Tables

**Figure 1 ijms-24-11503-f001:**
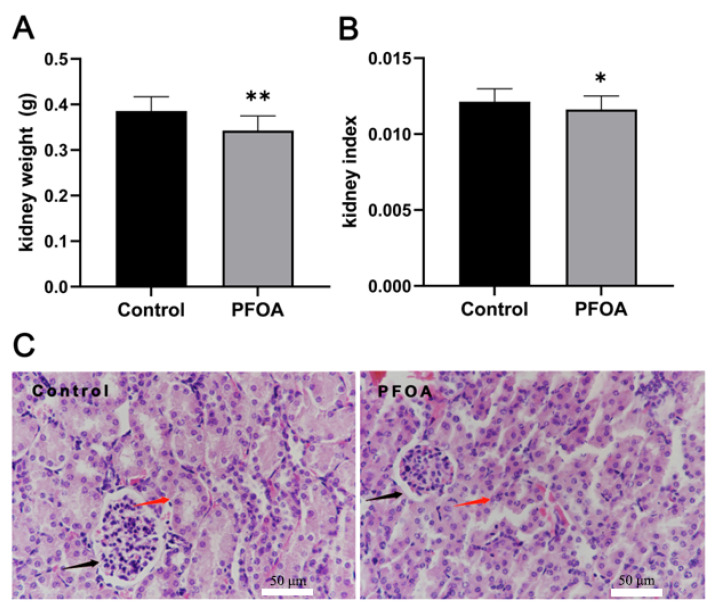
Effects of PFOA exposure during pregnancy on kidney weight, kidney index, and organ structure of offspring mice. (**A**) Kidney weight; (**B**) kidney index; (**C**) light micrographs (H&E, 400× magnification). Red arrows: renal tubules; black arrows: renal corpuscle. Data are presented as mean ± standard deviation (x ± SD). * *p* < 0.05, ** *p* < 0.01.

**Figure 2 ijms-24-11503-f002:**
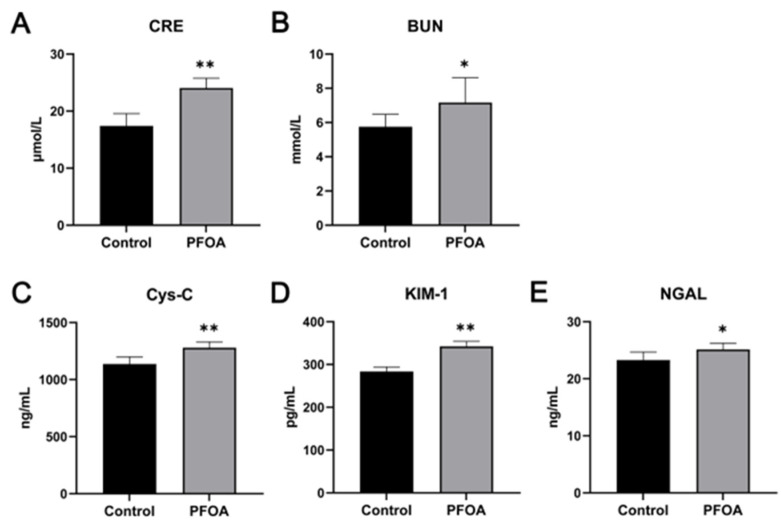
Effects of PFOA exposure during pregnancy on CRE and BUN levels, and markers of serum kidney injury. (**A**) Serum CRE; (**B**) serum BUN; (**C**) serum Cys-C; (**D**) serum KIM-1; (**E**) serum NGAL. Data are presented as mean ± standard deviation (x ± SD). * *p* < 0.05, ** *p* < 0.01.

**Figure 3 ijms-24-11503-f003:**
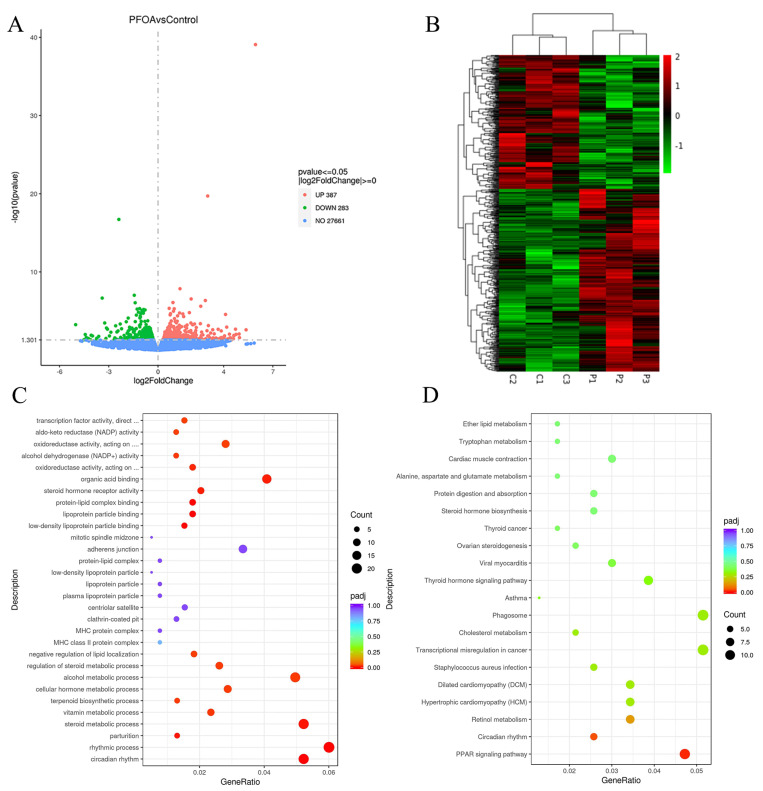
Transcriptomic analysis in kidneys of offspring after PFOA exposure during pregnancy. (**A**) Volcano plot of DEGs in kidneys; (**B**) heatmap of all DEGs between the control and PFOA group; (**C**) GO analysis of DEGs; (**D**) KEGG pathway enrichment.

**Figure 4 ijms-24-11503-f004:**
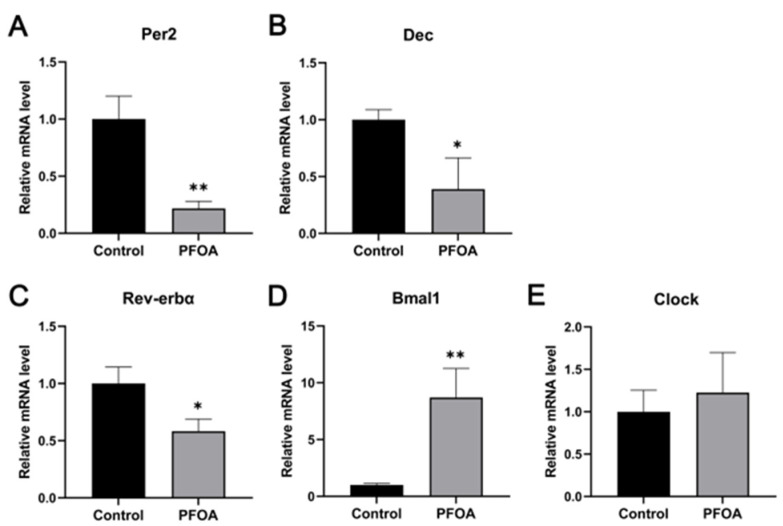
Relative mRNA levels of *per2* (**A**); *dec* (**B**); *rev-erbα* (**C**); *bmal1* (**D**); and *clock* (**E**). Data are presented as mean ± standard deviation (x ± SD). * *p* < 0.05, ** *p* < 0.01.

**Figure 5 ijms-24-11503-f005:**
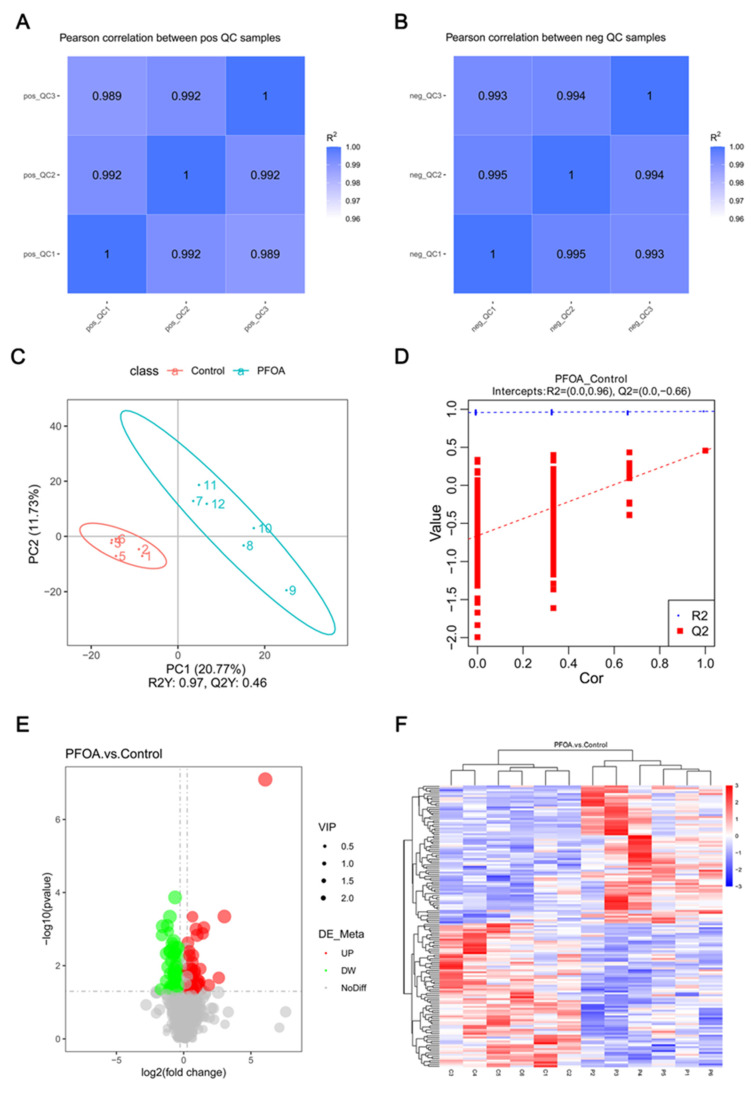
(**A**,**B**) Correlation analysis of QC samples; (**C**) metabolic profiles of kidneys depicted by PLS-DA scores plot; (**D**) sorting verification results; (**E**) volcano plot of differential metabolites in kidneys between the control and PFOA groups; (**F**) clustered heatmap of differential metabolites.

**Figure 6 ijms-24-11503-f006:**
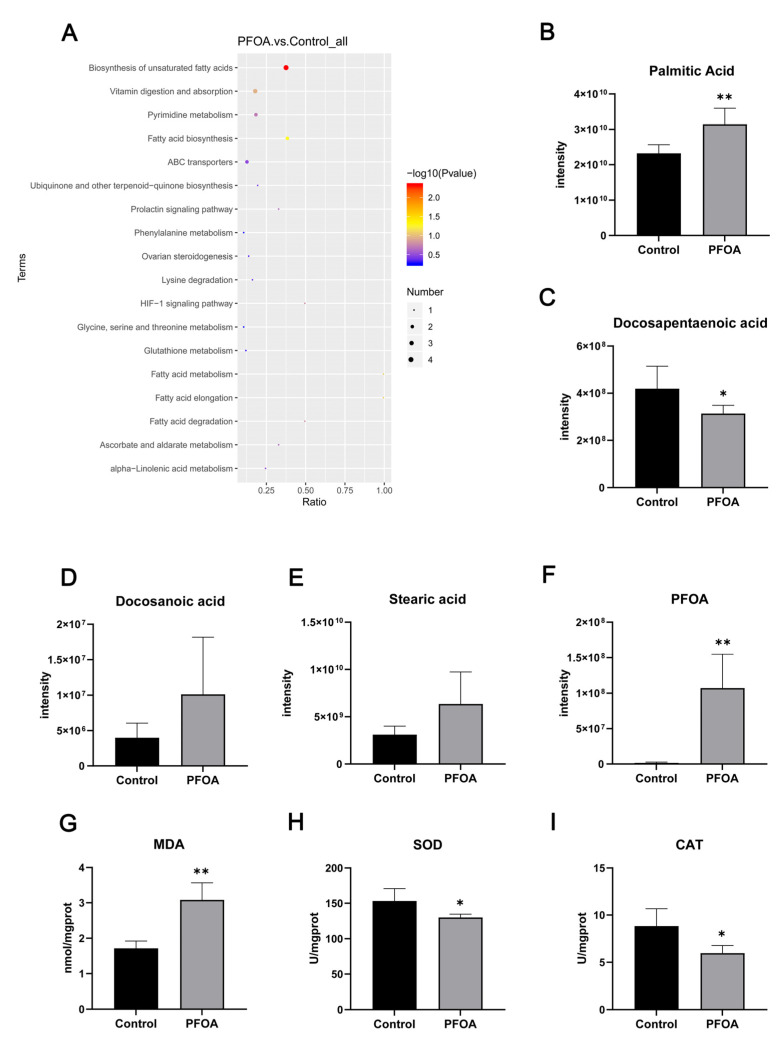
(**A**) Pathway analysis of the key renal metabolites; (**B**–**E**) relative levels of metabolites enriched in the biosynthesis of unsaturated fatty acids; (**F**) relative level of metabolites with the most significant difference. (**G**–**I**) Effects of PFOA exposure on renal oxidative stress. Data are presented as mean ± standard deviation (x ± SD). * *p* < 0.05, ** *p* < 0.01.

**Figure 7 ijms-24-11503-f007:**
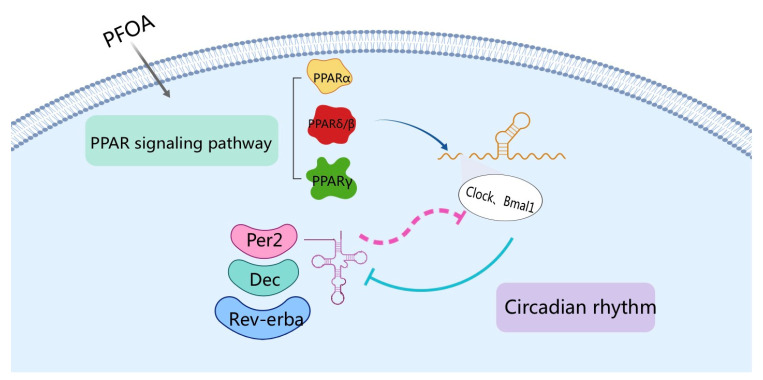
Relationship between PPAR signal pathway and circadian rhythm.

**Table 1 ijms-24-11503-t001:** Sequences of PCR primer pairs.

Gene Name		Primer Sequence (5′→3′)	GC%	Tm (°C)
*clock*	F	AACACAGCCAGCGATGTCTCA	52.4	64.3
	R	CATCCGTGTCCGCTGCTCTA	60.0	64.7
*bmal1*	F	AGGACTTCGCCTCTACCTGTTC	54.5	61.1
	R	ATAGCCTGTGCTGTGGATTGTG	50.0	62.4
*per2*	F	ATCAGCCATGTTGCCGTGTC	55.0	64.4
	R	CGTGCTCAGTGGCTGCTTTC	60.0	64.5
*rev-erb* *α*	F	GTGAAGACATGACGACCCTGGA	54.5	64.7
	R	GAGCCACTAGAGCCAATGTAGGTGA	52.0	65.0
*dec*	F	TGACATCAGATGACAGACTGGAG	47.8	60.3
	R	ACCCATGTCCCAAACTGGAG	55.0	62.1
*β-actin*	F	TCCTTCCTGGGCATGGAGT	57.9	63.0
	R	AGCACTGTGTTGGCGTACAG	55	60.0

## Data Availability

The data presented in this study are available on request from the corresponding author.

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
