# Peer review of "Transcriptome and Metabolome Analyses Reveal Perfluorooctanoic Acid-Induced Kidney Injury by Interfering with PPAR Signaling Pathway"

_ijms, 2023, doi:10.3390/ijms241411503_

Round 1
Reviewer 1 Report
This article provided research about the effects of perfluorooctanoic acid (PFOA) on offspring mice. PFOA has been a big concern worldwide since it has a long half-life and remains in water or soil for years, and our bodies accumulate it from intake. The authors performed transcriptome and metabolome analysis on pregnant mice exposed to PFOA. The study plan was good, and the results demonstrated in the article are clear and reasonable. Graphs are easy to interpret the influence of PFOA.
Their results will give additional insights into the current study on the effect of PFOA. However, the following points are suggestions to improve the manuscript to be publishable. In particular, there are a lot of grammatical errors. I recommend publishing after minor revisions.
- Please add Abbreviations including PFOS, Kim-1, NGAL (it was spelled out in Materials and Methods, though), MAD, CAT, SOD, etc. Although the authors spelled some words in the text, abbreviations in the article should be listed. It will help the reader to follow.
- Line 76, PFOS, appears for the first time. Since it is not a bio-related word, it is better to give the full name.
- There are many grammatical errors. Please see the English section. I recommend the authors proofread carefully.
- Line 260: with 72 and 75 up and down-regulated, ---> 72 up-regulated and 75 down-regulated
There are errors other than listed below. Please proofread it carefully.
Line 31. Per- and polyfluoroalkyl substances (PFAS) is a class of artificially synthesized organic fluorine compounds ïƒ ” is” must be “are.”
Line 33: hydrophobic and oleo-phobic ïƒ consistency: “Hydrophobic and oleophobic” or “Hydro-phobic and oleo-phobic”
Line 123; …. alcohol metabolic process and organic acid binding and other biological processes. --> alcohol metabolic process, organic acid binding, and other biological processes.
Line 169-171: Unnecessary capitalizations such as Biosynthesis of Unsaturated….
Line 170-171: fatty acids Acid biosynthesis ïƒ fatty acid biosynthesis
Line 243: 3 subtypesïƒ three subtypes (numbers less than ten should be spelled out)
Line 248: exhibt ïƒ exhibit
Reviewer 2 Report
In this study, the authors found the effect of PFOA on the biosynthesis of unsaturated fatty acids and the alteration of gene expressions including PPAR signaling pathway and Circadian rhythm. These are interesting finding, however, the functional relationship between PPAR signaling pathway and Circadian rhythm is still unclear. Although the authors cited several reports supporting their hypothesis for the functional crosstalk between PPAR signaling pathway and Circadian rhythm in discussion, it is somehow unsatisfactory. Of course, I could understand that the authors would attempt to clarify them in future research and their arrival point in the current research seems to be suitable for publishing in somewhere, therefore I would like to suggest the authors at least show the illustration explaining their hypothetical model in their study in revised version.
Reviewer 3 Report
Considering that even recently PFAS could be detected in the blood of the majority of American adults and consequently accumulated in body tissues with direct impact on the kidney function, and additionally, they were detected in water, air, soil etc., the present study could increase the current knowledge regarding the impact on the health and the modalities to improve the quality of life. The methodology was accurately and exhaustively presented and the results were adequately displayed. Furthermore, the conclusions were supported by the research findings. Hopefully, the results of your study will increase further attention of general population regarding the negative effects of PFAS, particularly perfluorooctanoic acid.
Author Response
Thank you for your approval of the article. We will conduct further research.